# Methods for improving participation rates in national self-administered web/mail surveys: Evidence from the United States

**Brady T. West** [ORCID] *, **Shiyu Zhang, James Wagner, Rebecca Gatward, Htay-Wah Saw, William G. Axinn** *

Survey Research Center, Institute for Social Research, University of Michigan-Ann Arbor, Ann Arbor, Michigan, United States of America

* bwest@umich.edu

## Abstract

In the United States, increasing access to the internet, the increasing costs of large-scale face-to-face data collections, and the general reluctance of the public to participate in intrusive in-person data collections all mean that new approaches to nationally representative surveys are urgently needed. The COVID-19 pandemic accelerated the need for faster, higher-quality alternatives to face-to-face data collection. These trends place a high priority on the evaluation of innovative web-based data collection methods that are convenient for the U.S. public and yield scientific information of high quality. The web mode is particularly appealing because it is relatively inexpensive, it is logistically flexible to implement, and it affords a high level of privacy and confidentiality when correctly implemented. With this study, we aimed to conduct a methodological evaluation of a sequential mixed-mode web/mail data collection protocol, including modular survey design concepts, which was implemented on a national probability sample in the U.S. in 2020–2021. We implemented randomized experiments to test theoretically-informed hypotheses that 1) the use of mail and increased incentives to follow up with households that did not respond to an invitation to complete a household screening questionnaire online would help to recruit different types of households; and 2) the use of modular survey design, which involves splitting a lengthy self-administered survey up into multiple parts that can be completed at a respondent's convenience, would improve survey completion rates. We find support for the use of mail and increased incentives to follow up with households that have not responded to a web-based screening questionnaire. We did not find support for the use of modular design in this context. Simple descriptive analyses also suggest that attempted telephone reminders may be helpful for the main survey.

## Introduction

As internet access has continued to spread in the United States, conducting surveys via the web has become an attractive option for scientific research on the general population to

**Data Availability Statement:** All de-identified data and code enabling replication of the statistical analyses presented in this revised manuscript can

be found in the GitHub public repository (https://github.com/bradytwest/AFHS).

**Funding:** This work was supported by the Eunice Kennedy Shriver National Institute of Child Health and Human Development (NICHD) of the National Institutes of Health (grant number R01HD095920; PI: B.T. West; website: https://www.nichd.nih.gov/). The funders had no role in study design, data collection and analysis, decision to publish, or preparation of the manuscript.

**Competing interests:** The authors have declared that no competing interests exist.

advance social science and policy evaluation. The primary advantages of web surveys include portability, flexibility, and confidentiality–web surveys allow respondents to complete surveys at whatever time and location is convenient and private for them. Collecting survey data via the web is also an attractive method for lowering data collection costs and curtailing data collection timeframes, both of which support greater scientific innovation [1–3]. These advantages extend when respondents are allowed to use multiple devices, including personal computers, laptops, tablets, and smartphones, further providing respondents with more options for convenience with little difference in measurement error between the devices [4–7]. This type of data collection approach is particularly attractive given the COVID-19 pandemic, which has strained face-to-face data collection operations in the U.S. and increased the need for the collection of population health data in a timely manner.

On the other hand, the advantages of the web mode can be offset by the serious disadvantages of non-universal access to the internet (and the coverage bias that it may introduce in survey research [8]), lower response rates compared to non-web modes [2, 9, 10], and the potential for nonresponse bias to mislead investigators. In practice, these issues can be remedied with sequential mixed-mode designs, where alternative non-web modes such as mail and telephone are used to follow up with non-respondents. This approach is informed by the leverage-salience theory of survey participation [11], where exclusive use of a single mode of data collection may not appeal to all individuals selected for a sample, and the use of alternative, complementary data collection protocols for nonresponse follow-up may help to recruit different types of respondents for whom the initial protocol was not appealing [12]. Sequential mixed-mode approaches have been shown to increase response rates while also possibly decreasing the nonresponse bias in survey estimates compared to single-mode surveys [1, 13–15].

## Guiding theoretical framework

Fortunately, while the general population's access to the internet in the U.S. has improved, the survey methodology of augmenting web surveys to provide a more robust representation of the general population has also advanced. Here we harness the key hypotheses from those advances to compare their ability to improve the population representation in surveys primarily implemented using the internet. First, we consider the initial household screening required to implement targeted (for example by age eligibility) surveys of individuals, building on what has been learned about the benefits of sequential mixed-mode designs and implementation of cash incentives. Second, we investigate the potential of modular designs that aim to lower the burden of participating in surveys and spread the participation effort across time to improve representation.

### The use of sequential mixed-mode approaches and cash incentives for household screening

Despite the clear potential of these sequential mixed-mode approaches, there is insufficient research evaluating the utility of using self-administered modes of data collection (e.g., web and mail) to measure complex, cross-sectional, national probability samples, where separate household screening and main survey data collection stages are necessary. Many national studies are beginning to transition to such a data collection approach, largely out of a need to reduce costs; see [13] for details. Household screening is often needed in such data collections to determine whether there are persons present in a sampled household who are eligible for a particular survey, where one or more eligible individuals in a household are then randomly selected and invited to complete the main survey by either web or paper. Very little existing literature has evaluated effective methods for increasing household response rates when sampled

households are invited to complete a screening questionnaire online. Those studies find general support for protocols grounded in both leverage-salience theory and social exchange theory [16]. In the latter case, the provision of incentives to sampled households or persons who are not intrinsically motivated by the topic of the survey may introduce extrinsic motivation to participate, along with feelings of a need to reciprocate if provided with a cash incentive by the data collector.

For example, the 2016 American National Election Study (ANES) included an internet sample in addition to a "standard" face-to-face sample. For the internet portion of the study, mail invitations including cash incentives (initially $10 or $20 prepaid as part of an experiment, and eventually escalating to promised offers of $40 and $80 for completion of a pre-election and post-election survey) were sent to a probability sample of addresses [17, 18]. Sampled households were initially invited to complete a brief screening questionnaire online, after which an adult in the household was randomly selected. If the household informant (i.e., the person supplying the information needed to randomly select one household member) was not selected, he or she was asked for additional information about the household and contact information for the selected respondent. If the informant was randomly selected, then the main survey started. The screening response rate was about 57%, which rivals the response rates seen today by many face-to-face surveys [19, 20]. See [17] for additional details.

Another study focused on a national survey of parents about the health of their children (the National Survey of Children's Health, or NSCH; see http://childhealthdata.org/learn/NSCH for details). This survey has been conducted using web/mail since 2016 by the U.S. Census Bureau. In 2016, the median survey length was 4.2 minutes for the screening survey and 26.5 minutes for the main survey, for those with children; these times were much smaller for those without children. The 2016 study also included a screener incentive experiment ($0, $2, $5). Screener response rates were 50.3%, 53.2% and 55.3% respectively, providing support for the use of higher incentives at the screening stage. They also experimentally evaluated the use of an additional cash incentive ($0, $2, $5, $10) in the third mailing for the main survey, finding that the use of additional (i.e., non-zero) incentives in nonresponse follow-up also helped to increase main response rates.

The National Household Education Surveys (NHES) obtained a 69% response rate for their screening survey in a large 2011 field test that involved converting from a telephone mode of data collection to mail (see [21–23] for details about this conversion, which improved coverage of the target population and increased the response rate for essentially the same cost). They tested $2 versus $5 incentives and found that the $2 incentive for the screener obtained a 67% response rate, and the $5 incentive did not increase the response rate significantly [24]. This study suggested that a $2 pre-paid incentive may be sufficient for invoking social exchange theory and improving response rates for self-administered screening questionnaires.

Prior to 2015, the Residential Energy Consumption Survey (RECS) was conducted entirely as a national face-to-face survey. In 2015, RECS implemented a web/mail survey in parallel with their face-to-face survey. As the RECS concerns features of the housing unit and not individuals, there is no screening operation to identify eligible persons, but we can still draw on their research to understand household decisions to complete self-administered questionnaires at the screening stage of a data collection. The implementation of the web/mail approach indicated that offering 1) the choice of web or mail completion, 2) a higher incentive (a $10 bonus) for web completion, and 3) mail follow-up using a brief, two-page questionnaire for nonrespondents, produced response rates between 54% and 56%, while only slightly increasing costs and maintaining the consistency of sample characteristics with American Community Survey benchmarks for the target population (despite lower response rates compared to the face-to-face data collection; [25–27]). The use of different protocols at different times provides

motivation for different types of participants, as suggested by leverage-salience theory. The 2020 RECS is now running exclusively as a self-administered web/mail survey because of these earlier studies.

Revisiting the NHES studies discussed earlier, they also found that a modest $5 incentive resulted in a sample that was more closely aligned with population benchmarks while also maintaining a response rate at the main stage that exceeded 70% [24]. This provides additional support for the ability of modest incentives to secure more balanced participation from different population subgroups in self-administered surveys.

These prior studies therefore suggest that 1) ANES screener completion rates benefited from incentives; 2) NSCH screener completion rates benefited from higher incentives in the context of a web/mail approach; and 3) the $2 incentive was just as effective as a $5 incentive for household screening in the NHES. We are therefore only aware of three studies that have provided support for these types of incentive strategies in a web-based screening approach applied to a national data collection, along with two studies (RECS and NHES) further supporting the use of incentives and a mixed-mode protocol to produce a more balanced sample when using strictly web/mail approaches for the main data collection. Collectively, we therefore have both theoretical and empirical support for the following hypothesis, in the specific context of a national study using mailed invitations to complete a screening questionnaire online: *The mailing of an additional unconditional incentive to respondents who still have not responded to the web and paper questionnaires will help to increase response rates and improve sample balance, relative to nonrespondents who do not receive a mailing with the additional incentives.*

## Modular design for the main survey

There is also a need to make the survey experience as convenient as possible for persons agreeing to complete the main self-administered survey. Lengthy web surveys may lead to survey breakoff [28, 29], where individuals start the main survey online but fail to complete it. Recent studies have suggested that respondents are not willing to spend lengthy amounts of time on web surveys, and especially those completed on mobile devices [30], making survey length and the burden of the response task very important considerations for designers of web surveys. *Modular survey design*, where a survey researcher explicitly divides a survey up into shorter parts (or "chunks") that a respondent can complete at different points in time, has been the subject of several recent methodological studies [30–36]. In theory, modular design would be expected to increase survey completion rates by reducing the *perceived burden* of a lengthy survey response task [31, 33, 34]. Innovative design strategies combined with the modular approach may also increase completion rates: for example, if participants are notified in advance that they will only need to answer a small number of questions at different points in time, or that they will receive a micro-incentive each time that they respond to a small set of questions (drawing on the cost-benefit theory of survey response; [16]), completion rates may increase overall [37, 38].

The emerging literature in this area has presented some evidence of positive benefits associated with the modular design approach. Initial experimental studies have suggested increased respondent convenience, increased initial rates of participation, and higher data quality among participating individuals [31, 33, 34]. Non-experimental studies in various international settings have generally echoed these findings [37, 39, 40]. However, other recent studies of this approach have produced mixed or negative findings, specifically with respect to overall survey *completion* rates and item nonresponse rates [31, 32, 34, 41, 42]. These studies have consistently called for more research into the lengths of survey modules and the appropriate amount of time between the modules.

Another recent national study that attempted to transition a general social survey (designed for a cross-sectional national sample) from the face-to-face mode to a self-administered web/mail mode is the German component of the 2017/2018 European Values Study (EVS). Wolf et al. [43] describe the results of a three-arm experiment demonstrating that a concurrent web/mail approach and matrix sampling (i.e., assigning random subsets of non-core questionnaire items to different subsamples; [44–47]) resulted in a viable and more cost-efficient alternative to face-to-face data collection, yielding shorter median interview lengths and fielding times (38 minute median interview length for the survey over 6–8 weeks, versus 59 minutes for the survey over 6 months) and higher response rates (36.1% vs. 28.0%) relative to the face-to-face approach. One outstanding issue based on the experiment was slightly worse representation of the general population by self-administered respondents, with the face-to-face approach yielding more younger and non-German respondents (among other slight differences relative to benchmarks). Positive results were also reported for similar transitions of the EVS in Denmark, Iceland, and Switzerland by [48].

We therefore see evidence in the literature of efforts to reduce survey length and respondent burden (via modular design or matrix sampling) being beneficial for participation in self-administered surveys. For additional evidence of the benefits of reducing survey length for participation in web and mail surveys, see [49–52]. To our knowledge, no studies have evaluated modular design as a tool for increasing survey completion rates when a sequential mixed-mode web/mail approach is used to measure a national sample. Given the evidence above, we present a second hypothesis: *A modular survey protocol will increase completion rates for a lengthy main web/mail survey, relative to individuals asked to complete the entire survey in a single sitting.* Although the literature is more mixed in this area, almost nothing is known about whether the effectiveness of the modular design approach varies across socio-demographic subgroups in this web/mail setting, which has important implications for future adaptive survey design strategies in this context. We seek to examine this question regarding variability of effectiveness across subgroups as part of our analysis as well. More lengthy national surveys are beginning to transition to the use of web/mail modes for data collection [13], and a test of this hypothesis will provide general information about the effectiveness of modular design for such surveys.

## Materials and methods

### Overview of the American Family Health Study

In April 2020, a new project known as the American Family Health Study (AFHS; see afhs.isr.umich.edu) initiated data collection with a national address-based probability sample of more than 19,000 U.S. addresses. We emphasize that the AFHS used an address-based probability sample design in an effort to produce unbiased national estimates of key population parameters. This distinguishes our approach from opt-in online surveys and volunteer surveys conducted on non-probability samples, which are frequently prone to various sources of selection and coverage bias that address-based probability samples can help to reduce [53]. The AFHS used a sequential mixed-mode mail/web protocol for push-to-web household (HH) screening to identify eligible persons aged 18–49. Selected eligible persons were then invited to complete a 60-minute web survey on reproductive health and family formation topics, using a second sequential mixed-mode mail/web protocol that encouraged the selected persons to respond to the "main" survey via the web. This design enabled testing of the first two hypotheses communicated above.

The questionnaire was translated from a face-to-face national survey: The National Survey of Family Growth (NSFG). A consent form describing the minimal risks and benefits of the

main study, the study procedures, and an assurance of confidentiality was presented either electronically (via the web) or on paper (for mail respondents) prior to the main survey instrument. Web respondents were asked to agree to the terms described before proceeding. Data collection continued until June 2021.

About half of eligible respondents selected from sampled households that completed the screening questionnaire were randomly assigned to the aforementioned "modular" condition, where the main survey data collection would involve completing three 20-minute surveys allowing for a two-week break before the next survey invitation. The remaining half was assigned to a "full" survey condition, where they were asked to complete the full 60-minute survey in a single sitting. Respondents in the full condition were able to take breaks and return to the survey where they left off. This aspect of the design enabled testing of the third hypothesis communicated above. Additional details regarding the AFHS sample design and data collection methodology can be found elsewhere (see https://afhs.isr.umich.edu/about-the-study/afhs-methodology/; [54]).

## AFHS screening protocol

The AFHS screening questionnaire was designed to collect a list of persons aged 18 years and over in the household. In the first phase of this protocol, we selected a stratified probability sample of addresses, oversampling addresses predicted to have an age-eligible (18–49 years old) person present and located in high-density minority areas. Sampled households received a mailed invitation (including a $2 cash incentive) addressed to the resident of a particular state, inviting an adult member of the household to complete a screening questionnaire online. In the second phase of this protocol, a follow-up reminder was sent one week after the mailed invitation in the form of a postcard.

In the third phase of screening, consistent with best practices [12], a follow-up mailing that included a paper version of the screening questionnaire was sent two weeks after the postcard. In the fourth phase of screening, 28 days after the initial invitation, a random subsample of 5,000 non-finalized sampled addresses was sent a priority mailing with a final invitation to complete the screening questionnaire and an additional $5 incentive. Information obtained from completed screening questionnaires was used to identify eligible persons within the sampled households. If there was only one eligible person in the household and this person completed the screening questionnaire online, then that person was immediately invited to complete the main AFHS survey online. If there was more than one eligible person in a household or the screening questionnaire was completed on paper, one eligible person was randomly selected at a later date by AFHS study staff and then invited by mail to complete the main survey online.

## AFHS main data collection protocol

Once an eligible respondent was randomly selected from a sampled household completing the screening questionnaire, an initial invitation to complete the main survey online was sent by mail to the selected respondent, and the letter promised a $70 token of appreciation once the completed survey was received (Phase 1). For the screener respondents who responded via web and were selected for the main survey, the initial invitation immediately followed the screening questionnaire (i.e., people were redirected to another web page starting the main survey). For all other cases, the initial invitation was scheduled 14 days after the screener questionnaire was completed. This initial invitation was followed two weeks later (Phase 2) by either a postcard or email reminder (if the selected respondent provided an email address). We did not send postcard reminders to individuals who elected to provide their email address

in the screening questionnaire, assuming that they would be comfortable receiving email reminders from the study; this also resulted in slight cost savings. Given the resources, we could have also sent postcards to those providing email addresses as an additional type of reminder, and future studies could evaluate such an approach.

In the third phase of the main data collection protocol, a further reminder was sent by email or text message (when these contact details were provided in the screening questionnaire) after three and five weeks. For eligible nonrespondents for whom we did not have an email or text-enabled phone number, we mailed a follow-up letter at four and six weeks that included a substantially shortened paper version of the questionnaire but still encouraged the respondent to complete the survey online. The six-week reminder was mailed in a USPS priority mailer. In the fourth and final phase of the main protocol (after six or seven weeks), our calling center staff made reminder telephone calls to nonrespondents with telephone numbers available from commercial data sources linked to our sampling frame or the initial screening questionnaire (83% of these nonrespondents had telephone numbers available–although some (12%) of these were found to be invalid during the reminder calls). These staff did not administer the survey upon making successful contact, but rather encouraged the nonrespondents to self-administer the survey and provided any information that would assist them in doing so. The staff left reminder voicemails if they were unable to make contact.

This main data collection protocol was repeated for *each survey module* for those selected respondents randomly assigned to the modular condition. These respondents were promised $20 for completing the first module, $20 for completing the second module, and $30 for completing the third module. Because of the complex skip logic and extensive dependence of later questions on answers to earlier questions in the original NSFG instrument, invitations to complete Module 2 were only sent to individuals who completed Module 1; those who did not complete Module 1 were invited to complete both Module 1 and Module 2 at the same time; invitations to complete Module 3 were only sent to individuals who completed Module 1 and Module 2; and those who completed Module 1 but did not complete Module 2 were invited to complete both Module 2 and Module 3 at the same time. Fig 1 summarizes the data collection protocols for the screening and main stages of the data collection.

In general, individuals from households that completed the screening questionnaire online also completed the main survey online (in the Full condition or the experimental modules): between 92.5% and 97.5% of these individuals completed the main survey online across the Full condition and the three modules. For households that completed the screening questionnaire on paper, between 54.9% and 69.6% of sampled individuals who completed the main survey completed it online (again across the Full condition and the three modules), meaning that the web was still preferred for the main survey among these households (although not as frequently).

For individuals participating in the main survey online, participant consent was obtained via an online survey interview consent form, either for the full survey or each individual module (depending on the experimental assignment). For individuals participating in the main survey by mail, a paper consent form was provided with each mailed invitation to complete the main survey, and participants were instructed to keep the consent form for their records. Both consent forms are available as part of the S1 Appendix.

All study procedures, materials, and methods were approved by the Human Subjects Institutional Review Board at the University of Michigan (Study HUM00167171).

**Screening Data Collection Protocol**

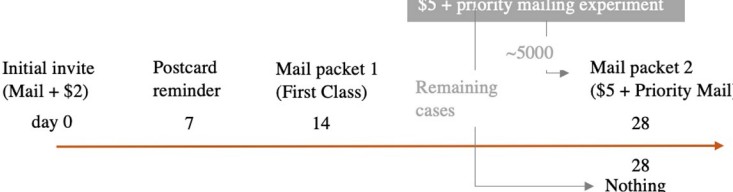

**Main Data Collection Protocol**

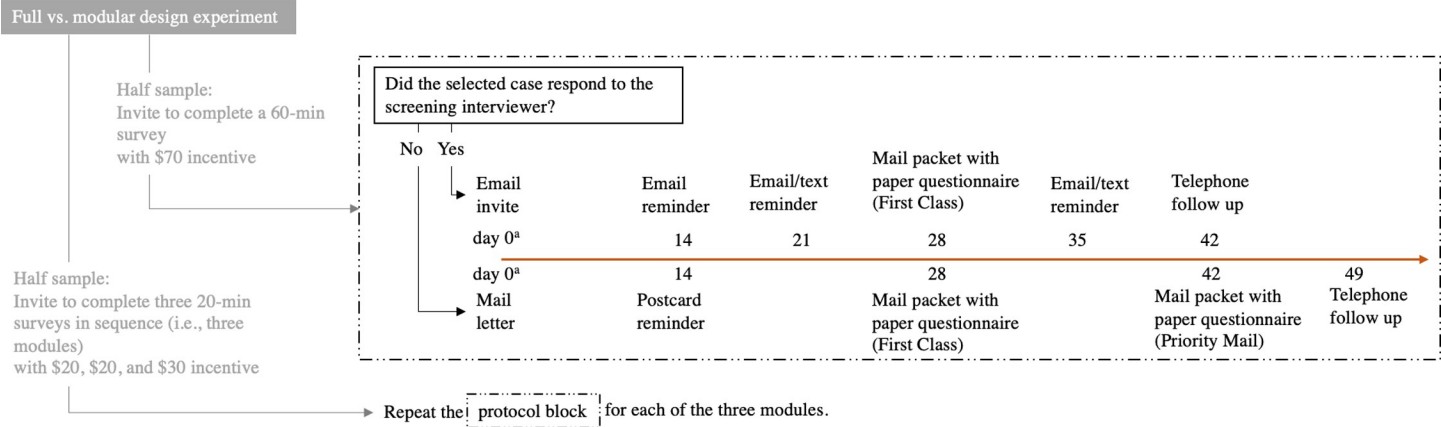

**Fig 1. Screening and main data collection protocols for the AFHS.**

## Data analysis

To test our first hypothesis (*The mailing of an additional unconditional incentive to respondents who still have not responded to the web and paper questionnaires will help to increase response rates and improve sample balance, relative to nonrespondents who do not receive a mailing with the additional incentives*), we first evaluated changes in the screener response rate across the four phases of the AFHS screening protocol. We considered the phase-specific response rates, cumulative response rates across the four phases, and response rates for each level of the $5 + priority experiment in the fourth phase. We also considered *hypothetical* cumulative response rates, assuming that *all* nonrespondents either did or did not receive the $5 + priority mailing follow-up in the fourth phase.

Next, in a series of exploratory analyses, we used Pearson chi-squared tests to evaluate the effects of the $5 + priority mailing follow-up at the fourth phase of the screening protocol on response rates among active nonrespondents, separately for each of 14 *LifeMode* groups of addresses defined by the *Esri Tapestry Segmentation* (https://www.esri.com/en-us/arcgis/products/data/data-portfolio/tapestry-segmentation) [55]. This system divides U.S. block groups into 14 distinct types of neighborhoods called *LifeModes* based on socio-demographic and socio-economic characteristics; similar commercial data describing different areas can be purchased and linked to sampling frames in other countries [56]. Each of the types is described psycho-graphically using terms like *Affluent Estates*. Although designed for marketing purposes, the Esri segments for the block groups, which can be linked to our full probability sample, have been found to be related to participation rates and mode choices in the 2015 Census

Test [57]. See Appendix I of the S1 Appendix for more details about this segmentation system, including analyses of the associations of the *LifeModes* with selected key variables in the AFHS.

With these supplemental analyses, we were exploring the possibility of *heterogeneity* in the effectiveness of the $5 + priority mailing intervention across these different types of areas. Assuming that the Esri Tapestry groups represent realistic descriptions of these areas, these types of exploratory analyses of heterogeneity in the effectiveness of data collection interventions are crucial for informing adaptive survey designs in future surveys that employ this kind of web/mail methodology, where protocols are tailored to the features of particular groups at the onset of a data collection to try to optimize the data collection outcomes. As part of these analyses, we also considered the mode of response (web or mail) in each of the experimental groups, to see if the $5 + priority mailing follow-up ultimately produced more web or mail (paper) responses to the screening questionnaire and whether these differences varied across areas defined by the 14 *LifeMode* groups.

To test our second hypothesis (*A modular survey protocol will increase completion rates for a lengthy main web/mail survey, relative to individuals asked to complete the entire survey in a single sitting*), we compared overall *completion rates* between the full and modular conditions at the main stage of data collection, where a selected person assigned to the modular condition needed to complete all three modules of the survey to be considered as a complete respondent. We also compared module-specific response rates to the overall response rate in the "full" condition. We then performed similar comparisons of the full and modular conditions for different subgroups defined by socio-demographic characteristics collected in the screening questionnaire, to examine whether the modular approach tended to appeal to specific socio-demographic subgroups.

All de-identified data files and code necessary to replicate the statistical analyses presented in this study are included with this article's S1 Appendix.

## Results

### AFHS response rates

We obtained an overall response rate in the screening stage of 15.0% and a conditional AAPOR RR4 response rate of 66.0% in the main stage (where for individuals randomly assigned to the modular condition, completing at least two sections of the questionnaire in the first 20-minute module was counted as a partial response). These two rates resulted in a net AAPOR RR4 response rate of 9.9%; we revisit this result in the Discussion. See Appendix II of the S1 Appendix for detailed descriptions of these response rate calculations.

Before examining the results relevant to our two hypotheses, we present some simple descriptive analyses suggesting that telephone contact attempts, made in an effort to remind individuals about the main survey for which they have been randomly selected, may help to improve main survey response rates. Table 1 suggests that attempts via telephone to remind initial nonrespondents with telephone numbers available about the main survey (or, about a specific module of the main survey), as part of the fourth and final phase of the main stage protocol, helped to increase the cumulative response rates in both experimental conditions. For the full condition and across all three modules in the modular condition, increases in the cumulative response rates from Phase 3 to Phase 4 of the main data collection stage ranged from 13.7 percentage points (Full) to 16.6 percentage points (Module 1). These increases were larger than all increases seen in previous phases.

We generally found that making telephone contact attempts was valuable during Phase 4 of the main data collection. While we were unable to contact the majority of the cases reaching Phase 4 without having responded, we found that response rates among those who were

**Table 1. Increases in cumulative response rates for the full sample assigned to each experimental condition (full vs. modular) across phases of the main data collection stage (where Phase 4 included the telephone reminders for individuals with telephone numbers available).**

| Exp. Condition | Sample Size[1] | Cumulative Response Rate After Phase 1 | Cumulative Response Rate After Phase 2 | Cumulative Response Rate After Phase 3 | Cumulative Response Rate After Phase 4 |
|---|---|---|---|---|---|
| Full | 766 | 41.0% | 46.2% | 52.9% | 66.6% |
| Modular | | | | | |
| Module 1 | 746 | 40.2% | 45.0% | 48.8% | 65.4% |
| Module 2 | 732 | 27.6% | 34.4% | 38.8% | 55.1% |
| Module 3 | 460 | 40.7% | 52.4% | 55.7% | 69.6% |

[1]Number of sampled individuals invited to: a) complete the full survey (Full) or Module 1 after a completed household screening questionnaire, b) complete Module 2 after completing Module 1, or complete Module 1 and Module 2 after initial nonresponse to Module 1 (Module 2), or c) complete Module 3 after completing Module 1 and Module 2, or complete Module 2 and Module 3 after initial nonresponse to Module 2 (Module 3).

contacted were much higher than among those who were not contacted. For example, in the Full condition, 22 out of the 50 cases reaching the telephone reminder phase that were successfully contacted ultimately responded (44.0%), while only 83 of the 304 cases that were not contacted ultimately responded (27.3%). For Module 1, successful contact resulted in response for 27 out of 43 cases (62.8%), while only 97 of the 331 cases that were not contacted responded (29.3%). Therefore, while we actually received more completed surveys from non-contacted cases in Phase 4 (which may have been due to the voicemail reminders, being reminded of the survey via caller ID, or delayed responses to the prior mailed reminders), efforts to make contact by telephone would seem to be important for increasing rates of response to the main survey.

These results, suggesting that the mail and telephone reminders used as part of the AFHS main stage data collection protocol may be helpful for increasing response rates in this self-administered web/mail context, are consistent with the findings of other recent studies. For example, a pilot study for a survey of saltwater anglers in two states in the U.S. found that varying the method of contact (including use of priority mailings) and following up with nonrespondents using telephone prompts/reminders tended to work well for increasing response rates to their main survey [58]. Other smaller studies conducted in specific locations have also consistently reported that telephone reminders tend to increase response rates among initial nonrespondents to a self-administered survey [59–61].

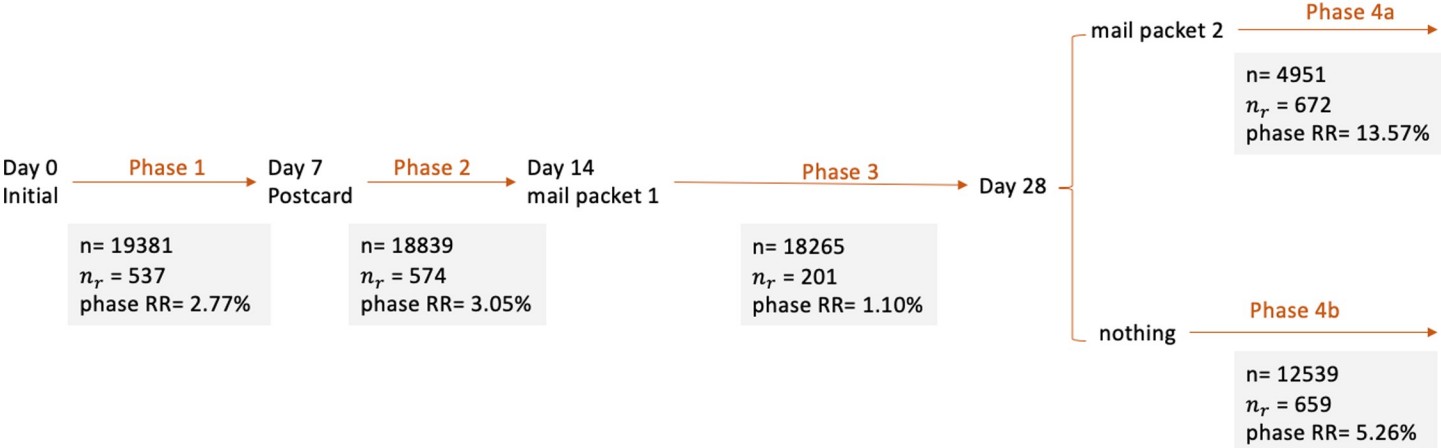

**Fig 2. Phase-specific response rates at the screening stage.**

Hypothesis 1: The mailing of an additional unconditional incentive to respondents who still have not responded to the web and paper questionnaires will help to increase response rates and improve sample balance.

Fig 2 presents the sample sizes for each phase of the AFHS screening protocol (n), obtained by subtracting the counts of respondents and nonrespondents from the available sample in the previous phase. For example, phase 1 had 537 respondents and 5 "hard" nonrespondents (who either called the main study telephone number or wrote a letter back to the study team with a hard refusal), and thus the sample size for phase 2 was equal to 18,839 (i.e., 19,381–537–5). Fig 2 also indicates the counts of respondents in each phase ($n_r$), and the corresponding raw phase-specific response rates across the phases of the AFHS screening protocol.

Fig 2 shows that the $5 + priority mailing follow-up in Phase 4 was highly effective in recruiting respondents (see [54] for more details). The response rate in Phase 4a is 8.32 percentage points (p.p.) higher than Phase 4b (where non-respondents did not get an additional $5 + priority follow-up).

Fig 3 presents a *cumulative* assessment of these results. Here, n indicates the overall sample size. The number 17,490 (in Phase 4a and 4b) indicates the number of sample cases entering the fourth phase; these cases were neither respondents nor "hard" nonrespondents in the previous three phases (corresponding to Fig 2 above, 17,490 = 4,951 + 12,539). For the fourth phase where the $5 + priority mailing experiment was conducted, we formed hypothetical scenarios where *all* Phase 4 cases either received (Phase 4a) or did not receive (Phase 4b) the $5 + priority mailing. Because these (hypothetical) response rates have the same denominator (i.e., the full sample), they can be directly subtracted from each other. Thus, the estimated effect of mail packet 1, assuming that all cases receiving mail packet 1 received no additional mailings (Phase 4b), is 5.78 p.p. (= 11.51% - 5.73%), and the estimated effect of the $5 + priority mailing (which would be sent *after* mail packet 1) is an additional 7.51 p.p. (= 19.02% - 11.51%). These results provide general support for our first hypothesis.

Wagner et al. [54] compared the random subsample of screening respondents that received the $5 + priority mail follow-up to the remaining "control" nonrespondents (who received no follow-up) with respect to their characteristics measured in the screening questionnaire. There were no differences in terms of screening respondents' sex, age, race and ethnicity, and household sizes. However, because only a portion of the screened households included an age-eligible person and the screening respondents might not be sampled for the main survey, Wagner and colleagues further compared the characteristics of the individuals that were actually selected to receive the main survey within eligible households. In this comparison, the experimental condition (the $5 + priority mail follow-up) selected a higher proportion of Black individuals to participate in the main interview than the control condition (where households received nothing). This in turn contributed to a greater number of Black respondents during the subsequent main survey stage. This helped to improve the diversity of the AFHS data since Black respondents were under-represented in the AFHS compared to the benchmark NSFG. Although prior literature has suggested that follow-up mailings can help with increasing respondent diversity in this way [62], we cannot determine how much of this increase was due to the priority mailing versus the additional $5.

In our exploratory analyses of potential heterogeneity in the effectiveness of the $5 + priority mailing across the Esri Tapestry *LifeMode* groups (Table 2), we found that this mailing did not significantly increase the response rate for the two *LifeMode* groups *Uptown Individuals* (3.26 p.p. difference between Phase 4a and 4b, $p = 0.208$) and *Scholars and Patriots* (5.17 p.p. difference, $p = 0.160$) (see Appendix I of the S1 Appendix for detailed descriptions of these groups). This mailing had the strongest effect in the *LifeMode* group *Rustic Outposts*, increasing its response rate by 11.89 p.p. ($p < 0.0001$).

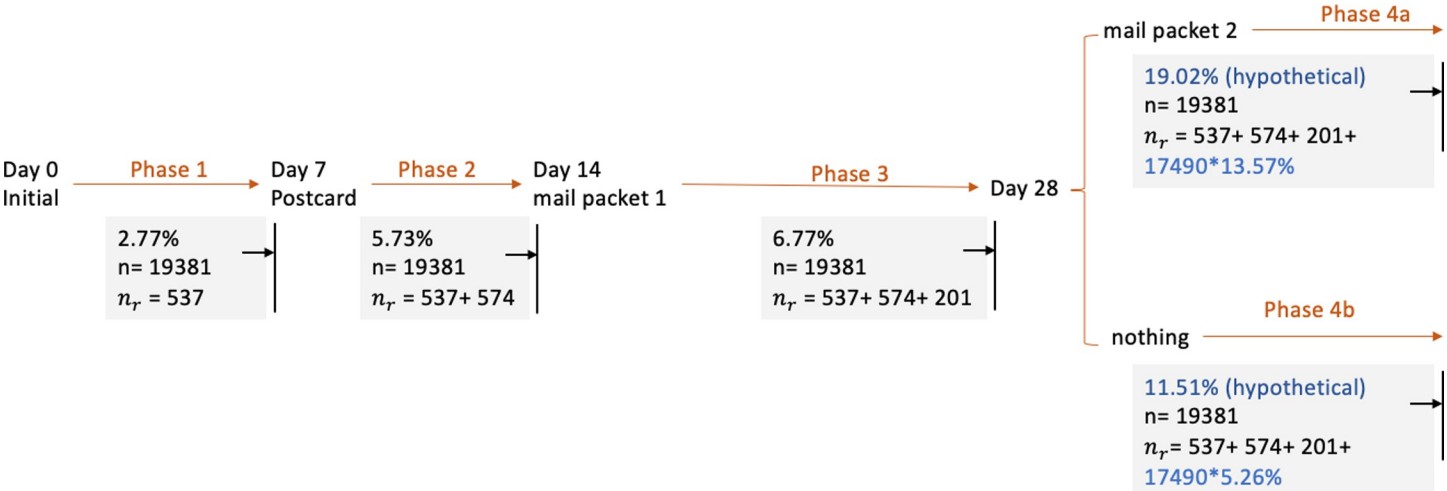

**Fig 3. Cumulative phase-specific response rates at the screening stage, in addition to hypothetical response rates at the conclusion of Phase 4 if all cases had either received or not received the $5 + priority mailing.**

Another way to understand the effect of the $5 + priority mailing is to break down the differences in response between the treatment and control groups by mode and then compare how these differences vary across Esri Tapestry *LifeMode* groups. On average, sending the additional mailing significantly increased the web response rate by 1.82 p.p. and the mail (paper) response rate by 6.50 p.p., relative to the control group (see the bottom of Table 2). The increases in web and paper responses were not consistent across Esri groups. Specifically, among *Uptown Individuals*, the web response rate was increased by a sizeable 4.44 p.p. ($p = 0.031$) but no effect was observed for the mail response rate (-1.18 p.p, $p = 0.553$),

**Table 2. Percentage point differences in screening response rates between Phase 4a and 4b by Esri *LifeMode* groups and by mode.**

| | Phase 4a | | | Phase 4b | | | Difference: Phase 4a –Phase 4b | | | | | |
|---|---|---|---|---|---|---|---|---|---|---|---|---|
| | Total | by Web | by PAPI | Total | by Web | by PAPI | Total | | by Web | | by PAPI | |
| *Affluent Estates* | 13.32 | 5.43 | 7.88 | 5.60 | 2.70 | 2.90 | 7.72 | *** | 2.73 | * | 4.98 | ** |
| *Upscale Avenues* | 12.99 | 3.94 | 9.06 | 5.15 | 3.03 | 2.12 | 7.84 | *** | 0.91 | NS | 6.94 | *** |
| *Uptown Individuals* | 10.45 | 8.46 | 1.99 | 7.19 | 4.02 | 3.17 | 3.26 | NS | 4.44 | * | -1.18 | NS |
| *Family Landscapes* | 14.50 | 6.39 | 8.11 | 4.80 | 2.40 | 2.40 | 9.70 | *** | 3.99 | ** | 5.71 | *** |
| *GenXurban* | 16.81 | 4.37 | 12.45 | 7.77 | 3.01 | 4.76 | 9.04 | *** | 1.36 | NS | 7.69 | *** |
| *Cozy Country Living* | 15.08 | 2.01 | 13.07 | 5.36 | 1.88 | 3.47 | 9.72 | *** | 0.13 | NS | 9.60 | ** |
| *Ethnic Enclave* | 9.66 | 2.62 | 7.04 | 4.00 | 2.08 | 1.92 | 5.66 | *** | 0.54 | NS | 5.12 | *** |
| *Middle Ground* | 14.69 | 4.90 | 9.79 | 4.88 | 1.61 | 3.28 | 9.81 | *** | 3.29 | *** | 6.51 | *** |
| *Senior Styles* | 14.86 | 2.25 | 12.61 | 6.50 | 1.91 | 4.59 | 8.36 | *** | 0.34 | NS | 8.02 | ** |
| *Rustic Outposts* | 16.37 | 2.68 | 13.69 | 4.48 | 1.49 | 2.99 | 11.89 | *** | 1.19 | NS | 10.70 | *** |
| *Midtown Singles* | 13.06 | 4.28 | 8.78 | 4.60 | 1.75 | 2.85 | 8.46 | *** | 2.53 | ** | 5.93 | *** |
| *Hometown* | 13.41 | 3.50 | 9.91 | 5.41 | 1.77 | 3.65 | 8.00 | *** | 1.73 | NS | 6.26 | *** |
| *Next Wave* | 9.97 | 2.66 | 7.31 | 3.88 | 2.87 | 1.01 | 6.09 | *** | -0.21 | NS | 6.30 | *** |
| *Scholars and Patriots* | 10.34 | 4.60 | 5.75 | 5.17 | 3.88 | 1.29 | 5.17 | NS | 0.72 | NS | 4.46 | NS |
| **Full Sample** | 13.57 | 4.08 | 9.49 | 5.26 | 2.26 | 2.99 | 8.32 | *** | 1.82 | *** | 6.50 | *** |

Note: * $p < 0.05$, ** $p < 0.01$, *** $p < 0.001$, NS = not significant. For the sake of space and readability, we do not provide detailed chi-square statistics and degrees of freedom, but this information is available upon request.

suggesting that the additional paper questionnaire might not be necessary for this specific subgroup. For *Scholars and Patriots*, the web and mail response rates were increased by 0.71 ($p = 1.00$) and 4.45 p.p. ($p = 0.06$), respectively, relative to the control group, suggesting that the additional follow-up had a small effect on this subgroup, regardless of mode. Finally, for *Rustic Outposts*, the strong effect of this mailing was largely because it increased the mail response rate by 10.7 p.p. ($p < 0.001$) relative to the control group. For these types of rural areas, the additional mail screening questionnaire combined with the additional incentive proved quite effective.

As for the remaining *LifeMode* groups, the \$5 + priority mailing always led to a significant increase in the mail response rate. But its effect on eliciting more web responses was less consistent. In half of these groups, the increase in the web response rate was small and non-significant. Thus, most of the screening questionnaires returned in response to the \$5 + priority mailing were on paper. In general, the effectiveness of the \$5 + priority mailing nonresponse follow-up did seem to vary across different types of geographic areas.

Hypothesis 2: Modular Survey Design Will Increase Main Survey Completion Rates.

We now evaluate whether the modular design approach increased response and completion rates overall. For the main instrument, the full approach resulted in a response rate of 62.9% (fully completed surveys), and 66.6% when including partial respondents. For the modular approach, only 42.9% of respondents completed all three modules with no partial responses ($p < 0.001$). Of those invited to complete module 1, 64.3% completed the module (not significantly higher than the full response rate overall), and this response rate was 65.4% when including partial respondents. Of the respondents who completed module 1, 82.6% completed module 2, and of those respondents, 79.2% went on to complete module 3. Given these results, the attrition in completing the entire survey that we found in the modular condition appeared to come from the additional requests to complete modules 2 and 3 (which is consistent with the prior literature in this area that was reviewed earlier). We therefore did not find support for our third hypothesis.

Next, we explored whether the modular approach tended to be more effective for particular types of individuals at the main stage. We found that the modular approach was particularly ineffective for non-Hispanic other males between the ages of 20–49, where the completion rate in the full condition was 69.3%, and the modular completion rate was 42.1% (including partials, $p < 0.001$). The same was true for Black and Hispanic males between the ages of 20–49, where the differences were 53.1% vs. 26.7% ($p < 0.05$) and 59.3% vs. 36.7% ($p < 0.05$), respectively (including partials). The modular approach simply did not seem to work for adult males. The same pattern held up for other (73.7% vs. 49.4%, $p < 0.001$) and Hispanic (58.9% vs. 38.0%, $p = 0.019$) females between the ages of 20–49.

One interesting difference was for Black females between the ages of 20–49. For this subgroup, the module 1 response rate (78.0%) was significantly higher than the full response rate (55.6%, $p < 0.05$). The percentage of individuals in this subgroup who completed all three modules (56.0%) was still slightly higher than the full response rate among those who completed the screening questionnaire, although this difference was not significant. This may be worth investigation in future adaptive designs. The module 1 difference did not hold up as significant when including partial respondents, but the difference was still notable. This was the only change in the pattern of results when including partial respondents.

## Discussion

### Summary of findings

At the *screening* stage of a national web/mail data collection, mailing a paper version of the screening questionnaire along with an additional incentive to individuals who had not yet replied to mailed invitations to complete the screening questionnaire online or on paper helped to increase screening response rates significantly. The effectiveness of this approach did tend to vary across different types of geographic areas, and it appeared to work especially well in rural areas like *Rustic Outposts*. For most types of areas (*Uptown Individuals* being an exception), the benefit of this approach arises from bringing in more screening responses by mail (rather than by web). As we noted earlier, attaching this type of commercial information to an address-based sampling frame is also relatively straightforward in different countries [56, 63]. Given these results, adaptive survey designs applying unique protocols to different subgroups defined by these commercial data warrant additional attention in the future.

A simple descriptive analysis suggested that attempted telephone reminders to complete the main survey for nonrespondents at the *main* stage of a national web/mail data collection with telephone numbers available (in this case at least six weeks out from the initial invitation to complete the main survey) may help to increase cumulative response rates. We found larger increases in cumulative response rates in the phase involving telephone reminders (if numbers were available) than seen in all prior phases of the main stage protocol. However, we emphasize that this should be interpreted as an upper bound on the effectiveness of this approach. Researchers cannot control who will provide a phone number (generally as a part of the screening stage) or who can be successfully contacted given that a phone number is available, so one can only evaluate the overall effectiveness of this approach for the total sample. Furthermore, there is no direct evidence of a counterfactual in this case. While the majority of the new responses in this phase of data collection came from individuals who were not contacted by telephone (and may have instead been responding to voicemail reminders or mailed reminders from prior phases), we did find that rates of response were much higher among individuals with whom we successfully made contact by telephone. This suggests that efforts to make contact by telephone with initial nonrespondents at the main stage of this type of self-administered web/mail data collection (for those with telephone numbers available) may be valuable, but this finding needs more careful experimental evaluation in future research.

With respect to the ability of modular survey design to increase perceived respondent convenience and reduce burden, we did not find support for our second hypothesis. Splitting a 60-minute web survey into three smaller modules (with completion times ranging between 5 and 20 minutes, depending on a respondent's characteristics) and offering a two-week break in-between the modules significantly reduced overall survey completion rates relative to asking the respondents to complete the survey online in one sitting and take breaks as needed. The reduction in the completion rate for the modular approach came primarily from lower response rates to the follow-up modules. Only for Black females aged 20–49 did this type of modular approach prove to be effective; reasons for this are unclear, and future research would benefit from more qualitative work to understand why this approach may be more effective for certain subgroups.

Overall, these findings are largely consistent with the relatively nascent literature in this area and provide more compelling evidence of the benefits of sequential mixed-mode design in these types of national web/mail surveys, where in-person interviewing is not an option for nonresponse follow-up and both mail and telephone follow-up efforts play important roles in increasing response rates and recruiting different types of respondents.

## Implications for practice

Regarding our main findings, we did see evidence of heterogeneity in the effectiveness of these approaches at the screening stage (where we had linked information from the Esri Tapestry Segmentation about the type of area where a sampled person was living). These findings for specific subgroups have important implications for future adaptive survey designs in these types of national web/mail data collections. For example, when following-up with non-responding sampled households in wealthier urban areas with younger adults, the $5 + priority mailing used as part of the screening stage may not need to include a paper screener (see also [42]). As another example, using a modular design approach for Black females between the ages of 20–49 at the main data collection stage may work well. However, these findings need replication in other settings. We found that linking the Esri Tapestry Segmentation data for our sampled areas to our sampling frame was a useful tool for evaluating areas where these approaches may have been more or less effective at the screening stage. These Esri segments are readily available, can easily be linked to address-based samples, and offer one way to cluster the distinctive features of neighborhoods based on data from 2010 Census, the American Community Survey, Esri's demographic updates and consumer surveys such as the Survey of the American Consumer ([55]; see also [56] for an example of this approach in another country). Future studies could use our results to tailor their screening approaches to particular types of areas defined by these data.

We did not find evidence of the modular data collection approach being effective at increasing response rates at the main data collection stage. We acknowledge that this approach was akin to asking respondents randomly assigned to this condition to complete three separate surveys, rather than a single survey (even though the modular surveys were one-third of the length, in an effort to decrease perceived burden). In short, this study suggested that asking respondents to complete several shorter web/mail surveys was not as effective as asking respondents to complete a single long web/mail survey and take breaks as needed; the number of survey requests, and not the length, seemed to be the limiting factor. Respondents may not have the time (or the regular privacy) required to complete several short surveys on potentially sensitive topics, thus preferring to finish an entire survey all at once. We do note that modular design may have other applications outside of breaking a survey into parts in an effort to reduce respondent burden and possibly increase response rates. For example, one may use a short initial survey to collect indicators that help to identify members of a population with certain characteristics, and then follow up with individuals having specific characteristics for additional data collection.

Overall, for future national web/mail surveys in the U.S. requiring household screening, we would recommend the following design strategies based on the results of this study:

1. mail nonrespondents to the initial web invitation a paper questionnaire and a small additional pre-paid incentive (e.g., $5);

2. when possible, attempt telephone reminders for nonrespondents with telephone numbers available at the main stage; and

3. allow respondents to complete the entire web survey in a single sitting (rather than splitting it into modules) and take breaks as needed.

Slight modifications of these overall suggested approaches may be possible based on our findings for particular subgroups, and replications in other settings (especially outside of the U.S. context) would be helpful for supporting these modifications more globally. Studies with more financial resources may also consider the use of face-to-face follow-up if the telephone

reminders do not prove effective; the AFHS did not have the resources to implement in-person interviewing at any stage of the study.

## Future research directions

First, our $5 + priority mailing approach used to follow up with nonrespondents at the screening stage did not enable decomposition of what was most effective: the $5, or the priority mailing? In a second national replicate of the AFHS that was recently completed, we attempted to decompose this effect with a designed experiment, and these analyses have indicated that the additional $5 and the priority mailer have *additive*, rather than *multiplicative*, positive effects on the response rate [64]. That is, the additional $5 incentive and the use of a priority-type mailer both serve to increase the response rate significantly, but these effects do not differ depending on whether the other technique is used simultaneously. Replications of this finding are needed in other contexts, especially in international settings.

Second, we still feel that further evaluation of the modular design approach in web surveys is worthwhile. The 20-minute survey request in this study may have negated some of the possible benefits of the modular approach, and shorter questionnaires combined with micro-incentives may still prove valuable for increasing perceived convenience [37]. Future experimental work with these kinds of approaches would be valuable, but this study suggests that the modular approach may not work well with longer, complicated surveys.

Third, our screening response rate was much lower than expected given the prior literature in this area and similar screening efforts from other national surveys prior to the COVID-19 pandemic. All the data collection for this replicate of the AFHS took place during the pandemic, and much of the data collection took place during the final year of the Trump administration. This context needs to be considered when evaluating the response rates, as a sensitive political climate combined with distrust in the government and/or science may have affected the response rates for this NIH-sponsored survey [65]. We will continue to evaluate response rates in our second national sample replicate of the AFHS, but ongoing reporting of response rates to these types of web/mail screening efforts in other contexts would be welcome from other survey organizations to see if these response rates were part of a larger societal pattern.

Finally, this paper did not address the effects of these alternative approaches on substantive responses to the main survey, or on other paradata related to the approaches (e.g., response times, item missing data, etc.). We are actively working on these evaluations, which generally provide support for the ability of the AFHS approach to replicate findings based on the NSFG despite the lower response rate [66], but future work needs to consider these substantive and paradata-related outcomes as well before the field can truly decide on optimal approaches for these types of national web/mail surveys.

## Supporting information

**S1 Appendix. Supporting information for "Methods for improving participation rates in national self-administered web/mail surveys: Evidence from the United States".**
(PDF)

## Acknowledgments

The authors wish to acknowledge the contributions of Mick Couper, Colette Keyser, Andrew Hupp, and the rest of the AFHS team in the Survey Research Operations unit of the Survey Research Center.

## Author Contributions

**Conceptualization:** Brady T. West, Shiyu Zhang, James Wagner, Rebecca Gatward, William G. Axinn.

**Formal analysis:** Brady T. West, Shiyu Zhang, Htay-Wah Saw.

**Funding acquisition:** Brady T. West, James Wagner, William G. Axinn.

**Investigation:** Brady T. West.

**Methodology:** Brady T. West, Shiyu Zhang, James Wagner, Rebecca Gatward, Htay-Wah Saw, William G. Axinn.

**Software:** Brady T. West, Shiyu Zhang.

**Supervision:** Brady T. West, Rebecca Gatward, William G. Axinn.

**Visualization:** Brady T. West, Shiyu Zhang.

**Writing – original draft:** Brady T. West, Shiyu Zhang.

**Writing – review & editing:** Brady T. West, Shiyu Zhang, James Wagner, Rebecca Gatward, Htay-Wah Saw, William G. Axinn.

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
