## [Decision Letter · Decision Letter 0]

13 Mar 2023

PONE-D-22-29437Methods for Improving the Quality of National Household SurveysPLOS ONE

Dear Dr. West,

Thank you for submitting your manuscript to PLOS ONE. After careful consideration, we feel that it has merit but does not fully meet PLOS ONE’s publication criteria as it currently stands. Therefore, we invite you to submit a revised version of the manuscript that addresses the points raised during the review process.

We look forward to receiving your revised manuscript.

Kind regards,

Bidhubhusan Mahapatra, Ph.D.

Academic Editor

PLOS ONE

“The authors wish to acknowledge the contributions of Mick Couper, Colette Keyser, Andrew Hupp, and the rest of the AFHS team in the Survey Research Operations unit of the Survey Research Center.”

“This work was supported by the Eunice Kennedy Shriver National Institute of Child Health and Human Development (NICHD) of the National Institutes of Health (grant number R01HD095920; PI: B.T. West; website: https://www.nichd.nih.gov/). The funders had no role in study design, data collection and analysis, decision to publish, or preparation of the manuscript.”

b) If there are no restrictions, please upload the minimal anonymized data set necessary to replicate your study findings as either Supporting Information files or to a stable, public repository and provide us with the relevant

URLs, DOIs, or accession numbers. For a list of acceptable repositories, please see http://journals.plos.org/plosone/s/data-availability#loc-recommended-repositories.

Additional Editor Comments:

It is very well written paper; however, as the reviewers highlight, there are some areas that needs more clarification.

Reviewers' comments:

Reviewer's Responses to Questions

**Comments to the Author**

1. Is the manuscript technically sound, and do the data support the conclusions?

Reviewer #1: Yes

Reviewer #2: Yes

Reviewer #3: Partly

2. Has the statistical analysis been performed appropriately and rigorously? 

Reviewer #1: Yes

Reviewer #2: Yes

Reviewer #3: Yes

3. Have the authors made all data underlying the findings in their manuscript fully available?

Reviewer #1: Yes

Reviewer #2: No

Reviewer #3: No

4. Is the manuscript presented in an intelligible fashion and written in standard English?

Reviewer #1: Yes

Reviewer #2: Yes

Reviewer #3: Yes

5. Review Comments to the Author

Reviewer #1: The issue undertaken in this manuscript entitled “Methods for improving the quality of national household surveys” is a highly relevant, novel, and timely initiative to improve the response rate from different population segments. These modes of data collection are more useful when the information is obtained on sensitive and private matters. However, these interventions to improve the response rate may not work or yield the same amount of impact across the globe settings. The reasons for such apprehensions are more practical and resources bound rather than theoretical. In a developing country setting, a face-to-face interview may have a better chance of a higher response rate to the national household surveys. It is because people still don’t trust outside communications (mail/Email) when it comes to divulging information regarding their households/dwelling units or any of its members. They trust only local representatives/leaders and their approval (oral/written) from them for providing any such information. Thus, face-to-face interaction between the interviewer/field supervisor and the local leader/representative regarding the aim and objective of the intended household survey is always helpful to reduce nonresponses in the initial phase of any household survey.

The authors on page 28 in the discussion regarding the modular approach stated-“In short, this study suggested that asking respondents to complete several shorter web/mail surveys was not as effective as asking respondents to complete a single long web/mail survey and take breaks as needed; the number of survey requests, and not the length, seemed to be the limiting factor”. However, the modular approach sometimes is also used to generate a specific indicator from the respondents possessing specific background characteristics, and not to divide a longer interview into serval shorter interviews. On the other hand, another support to your study could be that a typical kind of respondent wishes to finish the whole interview in one go as s/he may not get time/privacy to complete the survey again and again. In a face-to-face interview, such as conducted in many countries and varying settings under the Demographic and Health Survey (DHS), often respondents themselves requested to finish the interview at once. This is the reason for more than 85% of interviews in DHS get completed in the first visit itself. There could be various reasons for such a preference/pattern in a given setting.

Reviewer #2: I have fairly little to criticize on this paper. The paper is very nicely written, the research design is clear and capable to answer the research question. I have only some very general remarks, some of them can be easily incorporated, some others may produce doubts whether PLOS is the right outlet for this paper.

(1) The paper deals with a highly specific topic that primarily concerns practitioners of survey research. Frankly, I do not really believe that a paper that finds out that incentives and mail options increase response rates of population surveys is of interest for many readers. Thus, I was wondering whether this paper should be addressed to one of several survey methods journals. However, this is a question, the Editor needs to deal with. I look here only from the perspective of a survey methodologist.

(2) In front of your hypothesis 1, you state that there is "theoretical and empirical support for the first hypothesis" -- and I agree. However, it strikes me, why a hypothesis that is theoretical sound and well-supported by empirical evidence requires further research. This is also somewhat similar for the other hypotheses, but I agree that the empirical evidence is not as strong for those cases. Of course, I support the idea to replicate existing studies to see, whether the finding holds in different data, or whether the finding continues to be correct. It is, as you find out. But this then is again a research question that is highly specialized and not necessarily a topic for PLOS. In any case, you should try to explain why we need a further study for hypothesis 1.

(3) Given that this is a general audience outlet, I was wondering whether you should make crystal clear that you are dealing with the Online mode in a probability sample, and not with one of those self-selected Online Access Panels that gain so much attention on the public.

(4) The second hypothesis is already very specific. You expect an improvement of response rates for "motivated participants" on "mail or telephone reminders", which is fine, but raises again the question of general interest. The third hypothesis then is even more specific: It concerns only "carefully designed AND communicated modular surveys" with a "lengthy questionnaire". Can you extend a bit on whether it can be expected that these results transport to other survey, too?

(5) Hypothesis 3 mixes the hypothesis with the reasons for the hypothesis. I would separate the part behind "due to" from the hypothesis itself, especially since you are not investigating the "perceived convenience" mechanism.

(6) I was wondering if it is common for PLOS to print out so many differentiations of significance levels. This is meanwhile strongly discouraged by a number of other journals, and also by the ASA. I would also not present percentage points and percentage points differences with decimal places, but that's a matter of taste, perhaps.

(7) On pg. 20/21 you are reflecting on the differences in the result between the various ESRI groups. Doing so, you take the values at face value, stating that the hypothesis is true for one group, but not for some other. Another way to look at these inconsistent results is to state that the results are not robust at all, so that there is huge uncertainty whether the hypothesis is supported at all (despite the significance tests). There's also not really a theoretical reason why the hypothesis should be true for one group, but not the other. In fact, your theoretical reasoning suggest that it should be generally true.

(8) Maybe not your fault, but the resolution of the graphs was very bad in the file. Hard to read for my old eyes.

Reviewer #3: The title is far too broad and does not reflect the content of the paper. Please change it to something that reflects the experiment reported in the paper.

I find that the introductory sections of the paper (e.g. p.2, l.21-22; p.4, l.18-19) make unwarranted claims. The study tests just three particular aspects of survey design, and tests only a particular protocol for each aspect (other protocols are possible), so it should not be claimed that “A robust set of recommendations for new approaches to future surveys” can be provided as a consequence.

Overall, there are two robust tests included in this paper. One is the test of the effect of the $5+priority mailing and the other is the test of modular design. Both of these are based on carefully designed and implemented experiments. The rest of the paper consists of descriptive analysis accompanied by highly speculative interpretation. I strongly suggest that this analysis (which relates to hypothesis 1 and most of hypothesis 2) should be dropped and the paper more closely focussed on the two aspects that have robust design. Furthermore, the authors report on p.29 that a follow up study has separated the effects of the $5 and the priority mailing. I would strongly recommend combining that analysis with the analysis in this paper of the combined protocol, to provide readers with all the results on this specific part of the survey protocol in one place.

In the ‘Guiding theoretical framework’ section, all the examples given appear to be from one country. The authors should state whether they expect the framework and the study findings to be generalizable and, if so, to justify the restriction of examples to one country (or include examples from other countries). Alternatively, state clearly that this study applies to the U.S.

I find the hypotheses rather vague. For example, the first one (p.8, l.5-8) refers to the effect of several different components of design (sequential mixed-mode, mail options, cash incentives, web invitation by mailed letter). It would be much more informative to the reader if the effect of one or more of these components could be separated out. Furthermore, the counterfactual is not stated. The hypothesis is that this set of design components in combination “…will improve rates of response … along with …. quality …” But compared to what? Without knowing what we are comparing to, the hypothesis is meaningless.

The second hypothesis (p.9, l.1-3) has fewer design components confounded (“mail and telephone reminders”) but again refers to improving response rates. Compared to what? No reminders? That is not a very useful protocol to study as there is no good survey practice that involves no reminders. It has been clear since the 1950s that reminders are a cost-effective way to improve survey response rates.

The third hypothesis also fails to state a counter-factual, but this becomes apparent upon reading the design of the experiment.

Regarding the household screening, it is stated (p.13, l.19-20) that if there was only one eligible person, that person was immediately invited to complete the survey online. How was that achieved for households who completed the paper version of the screener? Was there an instruction with a web-link included within the screener questionnaire? How successful was this? It seems unlikely that someone filling a short paper questionnaire having declined the opportunity to do it online would then complete a long questionnaire online.

It appears (p.14, l.9-10) that postcard reminders were sent only to people who had not provided an email address. This seems odd, given the relatively low impact of emails (lots of people do not receive/see/open them). Why wouldn’t you send postcards to all, with the email additional for those with email addresses?

I don’t understand the claim (p.19,l.14) that mail packet 1 added 5.78 p.p. Fig.3 suggests 6.77 – 5.73 = 1.04 p.p. What am I missing?

p.20, l.4: “increase the diversity”: you have not presented evidence of this. To claim that increasing response amongst females (for example) increases diversity, you need also to show that females were under-represented in the absence of this feature (i.e. if you look only at web respondents). Also, it is unclear how you have used individual characteristics to classify households (this is household response).

The same comment about diversity applies to the analysis of the effect of the priority mailing (table 1): this does not tell us whether the priority mailing is improving the sample balance or making it worse.

The discussion of findings regarding hypothesis 2 (pp. 22-24) is rather unfocussed and reads a little as if data dredging is going on here. Lacking is a clear statement of which effects were tested and how many. Without this, the meaning of the differences reported cannot be judged. Including the results in table form could be helpful.

The reporting of hypothesis 3 findings suggests that invitation to module n+1 was conditional on completing module n. This was not clear in the Method section and requires clarification. Does it not make more sense to invite all screened-in people to each module?

p.28, l.4 “older Black females”: do you mean “younger Black females”?

The recommendations (p.29) are almost completely unsupported by the presented evidence. E.g. the authors recommend a push-to-web approach, despite the fact that they did not compare this to any other approach and only obtained a 15% response rate. Similarly, the use of telephone reminders was not explicitly compared to any alternative protocol. Recommendation #4 is the only one that is fully supported.

6. PLOS authors have the option to publish the peer review history of their article (what does this mean?). If published, this will include your full peer review and any attached files.

Reviewer #1: No

Reviewer #2: No

Reviewer #3: No

---

## [Author Response · Author response to Decision Letter 0]

25 Apr 2023

PLOS ONE Manuscript PONE-D-22-29437: Responses to Reviewer Comments

Note: All changes described below have been tracked in the “Revised Manuscript with Tracked Changes” file that has been submitted for this revision. We were thankful for the opportunity to resubmit this manuscript for further consideration.

Formatting Notes: The title page and the manuscript have been carefully formatted per PLOS ONE guidelines. Formatting changes have not been tracked in the clean submitted manuscript. We have also provided more details about the participant consent process in Materials and methods (and also in the supporting information) as requested. It was not clear whether we needed to provide additional information regarding our funding support, but we have left the statement of acknowledgments as-is. The funding statement indicated in the response letter is correct:

“This work was supported by the Eunice Kennedy Shriver National Institute of Child Health and Human Development (NICHD) of the National Institutes of Health (grant number R01HD095920; PI: B.T. West; website: https://www.nichd.nih.gov/). The funders had no role in study design, data collection and analysis, decision to publish, or preparation of the manuscript.”

We have also made all de-identified data and code used for the analyses available as part of the supporting information, enabling replication of the results; see https://github.com/bradytwest/AFHS.

Reviewer 1

COMMENT 1.1: The issue undertaken in this manuscript entitled “Methods for improving the quality of national household surveys” is a highly relevant, novel, and timely initiative to improve the response rate from different population segments. These modes of data collection are more useful when the information is obtained on sensitive and private matters. However, these interventions to improve the response rate may not work or yield the same amount of impact across the globe settings. The reasons for such apprehensions are more practical and resources bound rather than theoretical. In a developing country setting, a face-to-face interview may have a better chance of a higher response rate to the national household surveys. It is because people still don’t trust outside communications (mail/Email) when it comes to divulging information regarding their households/dwelling units or any of its members. They trust only local representatives/leaders and their approval (oral/written) from them for providing any such information. Thus, face-to-face interaction between the interviewer/field supervisor and the local leader/representative regarding the aim and objective of the intended household survey is always helpful to reduce nonresponses in the initial phase of any household survey.

RESPONSE 1.1: We agree for the most part with Reviewer 1 about this point. In many international settings, face-to-face surveys may be more effective and also less costly than in the U.S. Reviewer 3 provided similar comments. It became clear from these comments that we needed to do a better job of clarifying the scope of our work and the populations to which it would most readily apply. We have therefore made it clear, in the revised Abstract, the revised title of the manuscript, and the revised Introduction, that this work applies primarily to the U.S. context. 

COMMENT 1.2: The authors on page 28 in the discussion regarding the modular approach stated-“In short, this study suggested that asking respondents to complete several shorter web/mail surveys was not as effective as asking respondents to complete a single long web/mail survey and take breaks as needed; the number of survey requests, and not the length, seemed to be the limiting factor”. However, the modular approach sometimes is also used to generate a specific indicator from the respondents possessing specific background characteristics, and not to divide a longer interview into serval shorter interviews. On the other hand, another support to your study could be that a typical kind of respondent wishes to finish the whole interview in one go as s/he may not get time/privacy to complete the survey again and again. In a face-to-face interview, such as conducted in many countries and varying settings under the Demographic and Health Survey (DHS), often respondents themselves requested to finish the interview at once. This is the reason for more than 85% of interviews in DHS get completed in the first visit itself. There could be various reasons for such a preference/pattern in a given setting.

RESPONSE 1.2: We thank Reviewer 1 for these additional comments. We have expanded our discussion of this point where noted to make suggestions (along these lines) of other applications where modular design may be more effective / useful. We have also echoed the reviewer’s suggestion that many respondents may not have the time (or the required privacy) to complete multiple short surveys on sensitive topics, thus preferring to finish everything on one occasion.

Reviewer 2

COMMENT 2.1: I have fairly little to criticize on this paper. The paper is very nicely written, the research design is clear and capable to answer the research question. I have only some very general remarks, some of them can be easily incorporated, some others may produce doubts whether PLOS is the right outlet for this paper.

RESPONSE 2.1: We thank Reviewer 2 for the careful read and we appreciated this positive feedback. We have addressed all of these constructive comments in more detail below.

COMMENT 2.2: The paper deals with a highly specific topic that primarily concerns practitioners of survey research. Frankly, I do not really believe that a paper that finds out that incentives and mail options increase response rates of population surveys is of interest for many readers. Thus, I was wondering whether this paper should be addressed to one of several survey methods journals. However, this is a question, the Editor needs to deal with. I look here only from the perspective of a survey methodologist. 

RESPONSE 2.2: We appreciated this comment. The author team feels that the use of web surveys to study large populations has become a nearly ubiquitous tool in the social sciences; more and more, governments, large organizations, and businesses are attempting to measure large, widespread populations using web technologies. As a result, we aim to share this work in a general outlet — PLOS ONE – that will reach many researchers, practitioners, government and development agencies, and policy makers around the world who may be working on teams that employ such techniques for data collections. 

COMMENT 2.3: In front of your hypothesis 1, you state that there is "theoretical and empirical support for the first hypothesis" -- and I agree. However, it strikes me, why a hypothesis that is theoretical sound and well-supported by empirical evidence requires further research. This is also somewhat similar for the other hypotheses, but I agree that the empirical evidence is not as strong for those cases. Of course, I support the idea to replicate existing studies to see, whether the finding holds in different data, or whether the finding continues to be correct. It is, as you find out. But this then is again a research question that is highly specialized and not necessarily a topic for PLOS. In any case, you should try to explain why we need a further study for hypothesis 1.

RESPONSE 2.3: We note that the RECS and NHES examples immediately prior to the statement of our first hypothesis are not from screening operations for a national sample, but rather results from main survey data collections. Our first hypothesis primarily concerns screening operations, which have also become more common across the social sciences and applications of social science to specific population-scale studies. Our intent is to make the point that these two studies provide additional support for the type of mixed-mode, incentive-based protocol that we wanted to evaluate for *screening* a national sample. The prior three studies discussed before this suggested that 1) ANES screener completion rates benefited from incentives, 2) NSCH screener completion rates benefited from higher incentives and a web/mail approach, and 3) the $2 incentive was effective for the screener in NHES. We are therefore only aware of three studies that have provided support for these types of incentive strategies in a web-based screening approach applied to a national data collection, along with two studies supporting the use of incentives and a mixed-mode protocol to produce a more balanced sample when using strictly web/mail approaches for the main data collection. We therefore feel that more work is needed on the use of web/mail operations for household screening on a national scale as a result, and we have expanded our justification for further study in support of our first hypothesis accordingly where indicated. 

COMMENT 2.4: Given that this is a general audience outlet, I was wondering whether you should make crystal clear that you are dealing with the Online mode in a probability sample, and not with one of those self-selected Online Access Panels that gain so much attention on the public. 

RESPONSE 2.4: We appreciated this important point. We have added the following text (along with a new and relevant reference) to the first paragraph under Materials and methods, where we introduce the AFHS: “We emphasize that the AFHS used an address-based probability sample design in an effort to produce unbiased national estimates of key population parameters. This distinguishes our approach from opt-in online surveys and volunteer surveys conducted on non-probability samples, which are frequently prone to various sources of selection and coverage bias that address-based probability samples can help to reduce [63].”

COMMENT 2.5: The second hypothesis is already very specific. You expect an improvement of response rates for "motivated participants" on "mail or telephone reminders", which is fine, but raises again the question of general interest. The third hypothesis then is even more specific: It concerns only "carefully designed AND communicated modular surveys" with a "lengthy questionnaire". Can you extend a bit on whether it can be expected that these results transport to other survey, too?

RESPONSE 2.5: For each of these hypotheses, we have added text clarifying why testing these hypotheses would be of general interest for the large number of researchers conducting large-scale web/mail surveys. We removed the “motivated participants” language in the case of Hypothesis 2, as we feel that useful techniques for households that have already completed a screening questionnaire would be of general interest to researchers and practitioners using web/mail modes exclusively. We also removed the “carefully designed and communicated” language to make Hypothesis 3 more general.

COMMENT 2.6: Hypothesis 3 mixes the hypothesis with the reasons for the hypothesis. I would separate the part behind "due to" from the hypothesis itself, especially since you are not investigating the "perceived convenience" mechanism.

RESPONSE 2.6: Thank you for this point; we have modified the language for Hypothesis 3 accordingly.

COMMENT 2.7: I was wondering if it is common for PLOS to print out so many differentiations of significance levels. This is meanwhile strongly discouraged by a number of other journals, and also by the ASA. I would also not present percentage points and percentage points differences with decimal places, but that's a matter of taste, perhaps.

RESPONSE 2.7: We have removed the p < 0.10 significance notation, but we did retain the decimal places used in the original submission. We are happy to make additional changes if the Editor so desires.

COMMENT 2.8: On pg. 20/21 you are reflecting on the differences in the result between the various ESRI groups. Doing so, you take the values at face value, stating that the hypothesis is true for one group, but not for some other. Another way to look at these inconsistent results is to state that the results are not robust at all, so that there is huge uncertainty whether the hypothesis is supported at all (despite the significance tests). There's also not really a theoretical reason why the hypothesis should be true for one group, but not the other. In fact, your theoretical reasoning suggest that it should be generally true.

RESPONSE 2.8: With these analyses, we were attempting to explore the possibility of heterogeneity in the effectiveness of the $5 + priority mailing intervention across these different types of areas. We admit that we are assuming that the Esri Tapestry groups represent realistic descriptions of these areas. In response to this comment and other comments from Reviewer 3, we have more clearly framed these analyses as being exploratory and supplementary to the analyses related to our main hypothesis. These types of exploratory analyses of heterogeneity in the effectiveness of data collection interventions are crucial for informing adaptive survey designs in future surveys that employ this kind of web/mail methodology, where protocols are tailored to the features of particular groups at the onset of a data collection to try to optimize outcomes, and we felt that this would be of general interest to readers of this work. We have provided more motivation for these analyses (see pages 19-20 of the clean manuscript) and re-iterated this point when discussing these results (see page 31 of the clean manuscript).

COMMENT 2.9: Maybe not your fault, but the resolution of the graphs was very bad in the file. Hard to read for my old eyes.

RESPONSE 2.9: We apologize that these figures were difficult to read in the PDF version of the manuscript that was provided to reviewers. We did not hear anything about the image files submitted being incorrect, but we are happy to provide different versions of the image files to the Editor if there is in fact some type of problem with the clarity of the files.

Reviewer 3

GENERAL RESPONSE: We very much appreciated Reviewer 3’s critical review of the manuscript. We feel that the revisions made in response to the comments below have strengthened the manuscript significantly, and we would welcome any additional feedback or critiques based on the revision.

COMMENT 3.1: The title is far too broad and does not reflect the content of the paper. Please change it to something that reflects the experiment reported in the paper.

RESPONSE 3.1: We have changed the title of the manuscript to the following, which we believe better captures the scope and contributions of the paper: Methods for improving participation rates in national self-administered web/mail surveys: Evidence from the United States. 

COMMENT 3.2: I find that the introductory sections of the paper (e.g. p.2, l.21-22; p.4, l.18-19) make unwarranted claims. The study tests just three particular aspects of survey design, and tests only a particular protocol for each aspect (other protocols are possible), so it should not be claimed that “A robust set of recommendations for new approaches to future surveys” can be provided as a consequence.

RESPONSE 3.2: We have revised the language in the Abstract (p.2, l.21-22) and the Introduction (p.4, l.18-19) to be much more specific about the findings and contributions of the revised manuscript. In general, we have revised the language throughout the manuscript to make sure that we are not over-generalizing our findings or making claims of robustness that are not entirely supported by the analyses presented.

COMMENT 3.3: Overall, there are two robust tests included in this paper. One is the test of the effect of the $5+priority mailing and the other is the test of modular design. Both of these are based on carefully designed and implemented experiments. The rest of the paper consists of descriptive analysis accompanied by highly speculative interpretation. I strongly suggest that this analysis (which relates to hypothesis 1 and most of hypothesis 2) should be dropped and the paper more closely focussed on the two aspects that have robust design. 

RESPONSE 3.3: Our first hypothesis concerns the effect of the $5+priority mailing on screening response rates and the quality of the responding sample in terms of population representation (per comments from Reviewer 2, we have clarified why it is important for additional testing of this hypothesis). Reviewer 3 notes that this hypothesis is tested based on a robust experimental design. We note that we supplemented this overall evaluation of this approach with additional exploratory evaluations of the effectiveness of the approach for different types of socio-demographic areas. We have added language to the revised manuscript clarifying why these additional exploratory analyses of the effectiveness of the experimental approach across different types of areas is broadly important for researchers designing large-scale surveys using the mail/web approach exclusively; see RESPONSE 2.8 above. We have also dropped all speculative interpretations and simply reported the results of these exploratory analyses. We believe that they provide useful information that has the potential to spur future research into adaptive survey design.

 Per comments from Reviewer 2, we have also clarified the importance of Hypothesis 2 for informing future designs of strictly mail/web surveys. We acknowledge that the use of mail and telephone reminders was not implemented in an experimental fashion, and we could only attempt telephone reminders for households providing telephone numbers in the screening interview or with linked phone numbers available from the commercial MSG data (which has now been made clear in the revised manuscript). We also did not have design control over which main stage individuals chose not to respond to initial invitations to complete the main survey via the web (which resulted in mail follow-up surveys for all such individuals). Furthermore, we did not perform analyses of the effects of the mail and telephone reminders with respect to some type of control condition, which was a weakness of the original manuscript that we appreciated Reviewer 3 emphasizing. 

Because analyses related to Hypothesis 2 are important in this context, we have added new analyses to the revised manuscript that evaluate (in an observational fashion) what happened with initial nonresponding individuals at the main stage that did not have telephone numbers available or for whom we could not make contact, for contrast with those households that did have contact established (where reminders were provided). We believe that these new analyses (see the new Table 2), which also include multivariable modeling with adjustment for covariates (given the observational nature of the telephone reminder data), continue to provide strong support for the effectiveness of such a technique (the use of telephone reminders if web and mail modes do not work), and provide useful general information for survey researchers using strictly mail/web surveys. We emphasize the cost efficiency of such a reminder approach, given its demonstrated advantages in the revised manuscript. 

 Because it was not possible, given the design, to analyze completion rates for nonrespondents who did not receive the mail follow-up reminders (as everyone who had not responded at a certain point in the protocol received the reminder mailings), we made the decision to drop all analyses discussing the benefits of the mail follow-up from the revised manuscript, as there truly was no counterfactual in this case. We again thank Reviewer 3 for this important criticism. 

COMMENT 3.4: Furthermore, the authors report on p.29 that a follow up study has separated the effects of the $5 and the priority mailing. I would strongly recommend combining that analysis with the analysis in this paper of the combined protocol, to provide readers with all the results on this specific part of the survey protocol in one place.

RESPONSE 3.4: Given that the other study mentioned has now been accepted for publication, we now refer to the results of this follow-up study in the revised Discussion section. As we now note on page 33 of the clean revised manuscript, the follow-up study again demonstrated the effectiveness of the $5 and priority mail on motivating participation. It suggested that the effect of the $5 was stronger than that of priority mail, and the two effects are additive. We believe that these combined results provide a strong set of general recommendations related to this type of follow-up technique for screening households when using web and mail modes only.

COMMENT 3.5: In the ‘Guiding theoretical framework’ section, all the examples given appear to be from one country. The authors should state whether they expect the framework and the study findings to be generalizable and, if so, to justify the restriction of examples to one country (or include examples from other countries). Alternatively, state clearly that this study applies to the U.S.

RESPONSE 3.5: We appreciated this comment. In response, we have changed the title of the manuscript to refer to the U.S. context specifically, and made sure to clarify throughout the revised manuscript that our findings primarily apply to the U.S. context. In the revised Discussion, we have called for replications of this approach in other countries.

COMMENT 3.6: I find the hypotheses rather vague. For example, the first one (p.8, l.5-8) refers to the effect of several different components of design (sequential mixed-mode, mail options, cash incentives, web invitation by mailed letter). It would be much more informative to the reader if the effect of one or more of these components could be separated out. Furthermore, the counterfactual is not stated. The hypothesis is that this set of design components in combination “…will improve rates of response … along with …. quality …” But compared to what? Without knowing what we are comparing to, the hypothesis is meaningless.

RESPONSE 3.6: For each of our hypotheses, we have revised the language used to clearly describe expectations with regard to the components of the design being tested, and we have also clearly stated the counterfactual in each case (when possible, given the nature of the design). We also very much appreciated this comment about the need to have a clearly stated counterfactual. As noted above in RESPONSE 3.3, we have deleted all references to the effectiveness of the mail follow-up aspects of the protocols (given the complete lack of a counterfactual).

COMMENT 3.7: The second hypothesis (p.9, l.1-3) has fewer design components confounded (“mail and telephone reminders”) but again refers to improving response rates. Compared to what? No reminders? That is not a very useful protocol to study as there is no good survey practice that involves no reminders. It has been clear since the 1950s that reminders are a cost-effective way to improve survey response rates.

RESPONSE 3.7: This was a critical weakness of the initial submission, and we have added the analyses of the counterfactual described in RESPONSE 3.3 above as a result (see Table 2). We have also clarified that our focus is on the telephone reminders, and added the counterfactual to the statement of the hypothesis. We agree that this is good survey practice in general, but we specifically wanted to evaluate the impact of telephone reminders (as opposed to actual interviewing by telephone staff) as a cost-efficient approach to reminding screened participants about the main survey in this type of national web/mail data collection. We have also identified and cited a small number of other studies (of which we were previously unaware) that have provided additional evidence of this being an effective nonresponse follow-up technique in this self-administration context, and also emphasized their limitations to motivate the analyses being presented in this study (see page 10 of the clean manuscript).

COMMENT 3.8: The third hypothesis also fails to state a counter-factual, but this becomes apparent upon reading the design of the experiment.

RESPONSE 3.8: We have now clearly stated the counterfactual for Hypothesis 3.

COMMENT 3.9: Regarding the household screening, it is stated (p.13, l.19-20) that if there was only one eligible person, that person was immediately invited to complete the survey online. How was that achieved for households who completed the paper version of the screener? Was there an instruction with a web-link included within the screener questionnaire? How successful was this? It seems unlikely that someone filling a short paper questionnaire having declined the opportunity to do it online would then complete a long questionnaire online.

RESPONSE 3.9: In the revised manuscript, we have clarified how households completing the paper version of the screener were processed (see pages 15-16 of the clean manuscript); in short, these individuals did not receive an immediate invitation to complete the survey online, but instead received a follow-up mail invitation to complete the main survey. We did not provide a web link for the main survey in the screening questionnaire, because then we would lose control over whether the selected individual was actually the one completing the survey online. These households with a single eligible person were therefore given the same protocol as households completing the screener online, initially receiving an invitation to complete the main survey online (which would be more cost-efficient). We have now included summary statistics indicating the response modes for individuals completing screeners online and those completing screeners on paper (see page 18 of the clean manuscript); in fact, several individuals completing the screener on paper eventually completed the main survey (or the main survey modules) online (see page 18 of the clean manuscript).

COMMENT 3.10: It appears (p.14, l.9-10) that postcard reminders were sent only to people who had not provided an email address. This seems odd, given the relatively low impact of emails (lots of people do not receive/see/open them). Why wouldn’t you send postcards to all, with the email additional for those with email addresses?

RESPONSE 3.10: You are correct. We did only send postcard reminders to people who had not provided an email address. This assumed that email reminders would be much more effective for people who actually elected to provide their email address in the screening survey, and would also save some costs of mailing the additional postcards. We have noted how we could have used the alternative (and costlier) approach you describe where indicated (see page 16 of the clean manuscript).

COMMENT 3.11: I don’t understand the claim (p.19,l.14) that mail packet 1 added 5.78 p.p. Fig.3 suggests 6.77 – 5.73 = 1.04 p.p. What am I missing?

RESPONSE 3.11: The 11.51 – 5.73 = 5.78 calculation indicated here is based on the hypothetical calculated response rate (in Figure 3) as if no one had received the second mail package, and we did nothing else after sending the initial mail packet 1. We have added text clarifying this calculation where indicated (see page 23 of the revised clean manuscript).

COMMENT 3.12: p.20, l.4: “increase the diversity”: you have not presented evidence of this. To claim that increasing response amongst females (for example) increases diversity, you need also to show that females were under-represented in the absence of this feature (i.e. if you look only at web respondents). Also, it is unclear how you have used individual characteristics to classify households (this is household response).

RESPONSE 3.12: We have dropped this analysis from the revised paper, as recommended earlier by Reviewer 3, given that we did not have a counterfactual aspect of the mail follow-up design. See also RESPONSE 3.3.

COMMENT 3.13: The same comment about diversity applies to the analysis of the effect of the priority mailing (table 1): this does not tell us whether the priority mailing is improving the sample balance or making it worse.

RESPONSE 3.13: In a now-published study that we cite in the revised manuscript, Wagner et al. (2022) compared the random subsample of screening respondents that received the $5 + priority mail follow-up to the remaining “control” nonrespondents (who received no follow-up) with respect to their characteristics measured in the screening survey. There were no differences in terms of screening respondents’ sex, age, race and ethnicity, and household sizes. However, because only a portion of the screened households included an age-eligible person and the screening respondents might not be sampled for the main survey, Wagner and colleagues further compared the characteristics of the individuals that were actually selected to receive the main survey within eligible households. In this comparison, the experimental condition (the $5 + priority mail follow-up) selected a higher proportion of Black individuals to participate in the main interview than the control condition (where households received nothing). This in turn contributed to a greater number of Black respondents during the subsequent main survey stage. This helped to improve the diversity of the AFHS data since Black respondents were under-represented in the AFHS compared to a benchmark study (the National Survey of Family Growth, or NSFG). We have elaborated on these results on page 24 of the clean manuscript, and made it clear that we cannot exclusively attribute this increase in diversity to either the $5 or the priority mailing in this experimental design.

COMMENT 3.14: The discussion of findings regarding hypothesis 2 (pp. 22-24) is rather unfocussed and reads a little as if data dredging is going on here. Lacking is a clear statement of which effects were tested and how many. Without this, the meaning of the differences reported cannot be judged. Including the results in table form could be helpful.

RESPONSE 3.14: Given the new analysis conducted for the revised manuscript, we have prepared a new table showing the increase in response rates for those who did and did not get contacted at this phase of the main protocol (see Table 2). We have also refined our discussion of these results given the new analysis.

COMMENT 3.15: The reporting of hypothesis 3 findings suggests that invitation to module n+1 was conditional on completing module n. This was not clear in the Method section and requires clarification. Does it not make more sense to invite all screened-in people to each module?

RESPONSE 3.15: Because of the nature of the original NSFG instrument, many items in later modules depended on responses to items from previous modules. We therefore needed to make sure that invitations to subsequent modules were only provided to individuals who completed the previous modules. We have clarified this point in the Methods section (see page 17 of the clean manuscript).

COMMENT 3.16: p.28, l.4 “older Black females”: do you mean “younger Black females”?

RESPONSE 3.16: We have clarified that we mean black females between the ages of 20-49.

COMMENT 3.17: The recommendations (p.29) are almost completely unsupported by the presented evidence. E.g. the authors recommend a push-to-web approach, despite the fact that they did not compare this to any other approach and only obtained a 15% response rate. Similarly, the use of telephone reminders was not explicitly compared to any alternative protocol. Recommendation #4 is the only one that is fully supported.

RESPONSE 3.17: Reviewer 3’s points about the absence of counterfactuals have been well-taken, and we have made appropriate revisions as described above. Based on these changes and revisions, we have revised our recommendations for practice accordingly, and directly referred to the evidence in support of each recommendation from the revised manuscript (see page 32 of the clean manuscript). Again, we very much appreciated these critical comments, and we hope that the revision has sufficiently addressed these concerns.

---

## [Decision Letter · Decision Letter 1]

3 Jul 2023

PONE-D-22-29437R1Methods for improving participation rates in national self-administered web/mail surveys: Evidence from the United StatesPLOS ONE

Dear Dr. West,

Thank you for submitting your manuscript to PLOS ONE. After careful consideration, we feel that it has merit but does not fully meet PLOS ONE’s publication criteria as it currently stands. Therefore, we invite you to submit a revised version of the manuscript that addresses the points raised during the review process.

We look forward to receiving your revised manuscript.

Kind regards,

Bidhubhusan Mahapatra, Ph.D.

Academic Editor

PLOS ONE

Journal Requirements:

Reviewers' comments:

Reviewer's Responses to Questions

**Comments to the Author**

1. If the authors have adequately addressed your comments raised in a previous round of review and you feel that this manuscript is now acceptable for publication, you may indicate that here to bypass the “Comments to the Author” section, enter your conflict of interest statement in the “Confidential to Editor” section, and submit your "Accept" recommendation.

Reviewer #1: All comments have been addressed

Reviewer #2: All comments have been addressed

Reviewer #3: (No Response)

2. Is the manuscript technically sound, and do the data support the conclusions?

Reviewer #1: Yes

Reviewer #2: (No Response)

Reviewer #3: Partly

3. Has the statistical analysis been performed appropriately and rigorously? 

Reviewer #1: Yes

Reviewer #2: (No Response)

Reviewer #3: Yes

4. Have the authors made all data underlying the findings in their manuscript fully available?

Reviewer #1: Yes

Reviewer #2: (No Response)

Reviewer #3: Yes

5. Is the manuscript presented in an intelligible fashion and written in standard English?

Reviewer #1: Yes

Reviewer #2: (No Response)

Reviewer #3: Yes

6. Review Comments to the Author

Reviewer #1: I don't have any more comment on this manuscript. The revised version of the manuscript has come up very well. This would be a valuable knowledge addition to the science of the survey research method within and beyond USA.

Reviewer #2: (No Response)

Reviewer #3: I am glad to see that you have given careful consideration to all reviewer comments and in most cases have taken appropriate action that, overall, has greatly improved the paper in my view. However, one major concern remains. And I have a few additional comments related to material that you have now clarified or added.

My major concern is with what is now hypothesis 2. The way this hypothesis is tested does not appear to me to provide any useful information for informing future designs. This is because you test the effect of being successfully contacted. This is not a design feature that can be controlled: the survey researcher cannot choose which sample members will be successfully contacted. Rather, this is an outcome of the process. I would strongly prefer you drop this hypothesis and analysis from the paper: I think it severely weakens the paper and does not give a good impression of the authors (unlike the analyses of the other two hypotheses, which are underpinned by robust experimentation). Alternatively, you could just include a descriptive analysis (estimate) of the overall effect of the telephone reminders for the whole sample. In practice, this is the treatment that researchers can implement: they cannot control who will provide a phone number or who can be successfully contacted, so the outcome for the total sample is the metric of interest. Of course, you have no direct evidence of the counterfactual, so you can only present as a maximum bound on the effect the difference between to total observed response and the total response if you remove all who responded subsequent to a successful phone contact (true effect is probably a little smaller as some of those contacted by phone may have eventually participated anyway).

Other comments:

p.7, l.11-12: I don’t understand this. If screener completion rate was 53.0% and weighted completion rate for main survey was 33.0%, how is the overall rate 40.7%? Surely it’s around 0.53 * 0.33 = 0.175 (if you assume screener response rate to be independent of eligibility for main stage). What am I missing?

p.9, l. 5-6: I still don’t see any evidence that completion rates have benefitted from a web/mail approach? This doesn’t seem to have been compared with any other approach? Maybe you mean that they have benefitted from incentives in the context of a web/mail approach?

p.9, l.6: “$2 incentive was effective”: reader could be excused for assuming this means relative to no incentive. But actually the evidence you presented was that $2 was as effective as $5.

p.17, l.17-19: thank you for attempting to clarify this. However, this information contradicts the notes to figure 1 (“If invited cases did not respond to mod 1, the later invitation would invite them to complete mods 1 and 2 together” and “Nonresponse to earlier modules resulted in continued invitation”). So this still requires clarification!

p.18, l.3-9: This text is written as if it is always the same person completing screener and main questionnaire, which is presumably not the case?

p.21, l.7-11: Related to my main concern above, this adjustment – upon which your estimation relies – is likely to be woefully inadequate. I doubt these variables account for more than a small fraction of the variation in propensity to respond to a survey. So, what remains is that more willing people turn out to be more willing….

7. PLOS authors have the option to publish the peer review history of their article (what does this mean?). If published, this will include your full peer review and any attached files.

Reviewer #1: No

Reviewer #2: No

Reviewer #3: No

---

## [Author Response · Author response to Decision Letter 1]

14 Jul 2023

Please see our attached file "Response to Reviewers.docx" for our detailed responses to the remaining reviewer comments.

---

## [Decision Letter · Decision Letter 2]

25 Jul 2023

Methods for improving participation rates in national self-administered web/mail surveys: Evidence from the United States

PONE-D-22-29437R2

Dear Dr. West,

We’re pleased to inform you that your manuscript has been judged scientifically suitable for publication and will be formally accepted for publication once it meets all outstanding technical requirements.

Kind regards,

Bidhubhusan Mahapatra, Ph.D.

Academic Editor

PLOS ONE

Additional Editor Comments (optional):

Reviewers' comments:

Reviewer's Responses to Questions

**Comments to the Author**

1. If the authors have adequately addressed your comments raised in a previous round of review and you feel that this manuscript is now acceptable for publication, you may indicate that here to bypass the “Comments to the Author” section, enter your conflict of interest statement in the “Confidential to Editor” section, and submit your "Accept" recommendation.

Reviewer #3: (No Response)

2. Is the manuscript technically sound, and do the data support the conclusions?

Reviewer #3: Yes

3. Has the statistical analysis been performed appropriately and rigorously? 

Reviewer #3: Yes

4. Have the authors made all data underlying the findings in their manuscript fully available?

Reviewer #3: Yes

5. Is the manuscript presented in an intelligible fashion and written in standard English?

Reviewer #3: Yes

6. Review Comments to the Author

Reviewer #3: I am really very grateful that you have taken all my comments - possibly a little demanding - in good spirit and have carefully amended the manuscript in the ways suggested or implied. I truly think the paper is much improved as a result and that the revisions were worthwhile. I hope you do too.

7. PLOS authors have the option to publish the peer review history of their article (what does this mean?). If published, this will include your full peer review and any attached files.

Reviewer #3: **Yes: **Peter Lynn

---

## [Editor Report · Acceptance letter]

27 Jul 2023

PONE-D-22-29437R2 

Methods for improving participation rates in national self-administered web/mail surveys: Evidence from the United States 

Dear Dr. West:

I'm pleased to inform you that your manuscript has been deemed suitable for publication in PLOS ONE. Congratulations! Your manuscript is now with our production department. 

Kind regards, 

on behalf of

Dr. Bidhubhusan Mahapatra 

Academic Editor

PLOS ONE